# Effects of a High-Protein Diet on Kidney Injury under Conditions of Non-CKD or CKD in Mice

**DOI:** 10.3390/ijms24097778

**Published:** 2023-04-24

**Authors:** Shohei Tanaka, Hiromichi Wakui, Kengo Azushima, Shunichiro Tsukamoto, Takahiro Yamaji, Shingo Urate, Toru Suzuki, Eriko Abe, Shinya Taguchi, Takayuki Yamada, Ryu Kobayashi, Tomohiko Kanaoka, Daisuke Kamimura, Sho Kinguchi, Masahito Takiguchi, Kengo Funakoshi, Akio Yamashita, Tomoaki Ishigami, Kouichi Tamura

**Affiliations:** 1Department of Medical Science and Cardiorenal Medicine, Graduate School of Medicine, Yokohama City University, 3-9 Fukuura, Kanazawa-ku, Yokohama 236-0004, Japan; 2Cardiovascular and Metabolic Disorders Program, Duke-NUS Medical School, 8 College Road, Singapore 169857, Singapore; 3Renal-Electrolyte Division, Department of Medicine, University of Pittsburgh, 3550 Terrace Street, Pittsburgh, PA 15261, USA; 4Department of Neuroanatomy, School of Medicine, Yokohama City University, 3-9 Fukuura, Kanazawa-ku, Yokohama 236-0004, Japan; 5Department of Investigative Medicine, Graduate School of Medicine, University of the Ryukyus, 207 Uehara, Nishiharacho, Okinawa 903-0215, Japan

**Keywords:** chronic kidney disease, remnant kidney model, 129/Sv mouse, high-protein diet, glomerular injury, tubulointerstitial injury

## Abstract

Considering the prevalence of obesity and global aging, the consumption of a high-protein diet (HPD) may be advantageous. However, an HPD aggravates kidney dysfunction in patients with chronic kidney disease (CKD). Moreover, the effects of an HPD on kidney function in healthy individuals are controversial. In this study, we employed a remnant kidney mouse model as a CKD model and aimed to evaluate the effects of an HPD on kidney injury under conditions of non-CKD and CKD. Mice were divided into four groups: a sham surgery (sham) + normal diet (ND) group, a sham + HPD group, a 5/6 nephrectomy (Nx) + ND group and a 5/6 Nx + HPD group. Blood pressure, kidney function and kidney tissue injury were compared after 12 weeks of diet loading among the four groups. The 5/6 Nx groups displayed blood pressure elevation, kidney function decline, glomerular injury and tubular injury compared with the sham groups. Furthermore, an HPD exacerbated glomerular injury only in the 5/6 Nx group; however, an HPD did not cause kidney injury in the sham group. Clinical application of these results suggests that patients with CKD should follow a protein-restricted diet to prevent the exacerbation of kidney injury, while healthy individuals can maintain an HPD without worrying about the adverse effects.

## 1. Introduction

Adequate consumption of dietary protein is critical for the maintenance of optimal health during normal growth and aging. One of the effects of a high-protein diet (HPD) intake is to promote muscle protein synthesis [1]. This effect inhibits the loss of skeletal muscle mass due to sarcopenia in the elderly [2,3]. Furthermore, HPD intake positively affects calcium and bone homeostasis through its effects on calcium absorption and bone turnover [4,5]. A significant positive association between protein intake and bone mineral density has been shown by epidemiologic studies [6]. HPD intake could inhibit age-related bone loss and decrease fracture risk due to osteoporosis [7,8]. In addition, HPD intake has effects of lower daily energy intake through increased satiety [9] and higher daily energy expenditure through the increased thermic effect [10]. Therefore, HPD intake decreases body weight and maintains lean body mass in obese patients [11,12]. Furthermore, protein intake regulates endogenous glucose metabolism and insulin sensitivity [13,14]. It has been shown that HPD intake can stabilize blood glucose and lower postabsorptive and postprandial insulin secretion during weight loss [15]. Cross-sectional clinical studies have shown that dairy protein consumption is inversely related to the incidence of impaired fasting glycemia and type-two diabetes [16,17]. Considering the prevalence of overweight individuals, obesity and global aging, the consumption of an HPD may be advantageous.

Although an HPD has many benefits, we must also consider the effects of a high-protein diet on chronic kidney disease (CKD). The number of patients with CKD continues to increase worldwide. The progression of CKD culminates with end-stage kidney disease, which eventually necessitates kidney replacement therapy [18,19,20]. Patients with CKD are at a high risk of mortality due to the high incidence of cardiovascular disease that is associated with kidney disease [21,22]. In 2017, 697.5 million cases of CKD were reported worldwide, with 1.2 million people ultimately dying from the disease [23]. Therefore, the development of interventions that prevent the progression of CKD represents a global challenge.

HPD intake aggravates kidney dysfunction in patients with chronic kidney disease (CKD) [24,25]. HPD intake results in the dilation of afferent arterioles in the glomeruli, which increases intraglomerular pressure and leads to glomerular hyperfiltration [26]. Long-term glomerular hyperfiltration leads to glomerulosclerosis, hypoxia in the tubulointerstitium and consequent tubulointerstitial inflammation and fibrosis [27]. In contrast, the effects of HPD intake on kidney function in healthy individuals are controversial. In one long-term observational study, no association between increased protein intake and decreased estimated glomerular filtration rate (eGFR) was observed in the group with normal renal function [24]. Alternatively, in the Gubbio study, higher protein intake was associated with lower eGFR in the group with normal renal function [25]. In the present study, we tested the hypothesis that HPD intake would have adverse effects on CKD kidneys but no apparent detrimental effects on healthy kidneys in mice.

CKD is diagnosed based on a low glomerular filtration rate (GFR; <60 mL/min/1.73 m^2^) and/or findings indicative of kidney injury, such as proteinuria, that have persisted for more than three months [28]. Numerous animal models have been developed over the years to elucidate the mechanisms involved in the pathogenesis of CKD and the effects of interventions [29]. The remnant kidney model, created by performing a 5/6 nephrectomy (Nx), is characterized by a low GFR with overt albuminuria; therefore, it has been widely used in CKD research. The mechanism associated with kidney injury in the remnant kidney model involves an increase in single-nephron GFR within the remaining glomeruli to compensate for nephron loss [30]. The subsequent increase in intraglomerular pressure leads to intraglomerular hypertension, and the resulting barotrauma induces glomerulosclerosis and proteinuria [31].

In this study, we employed a remnant kidney mouse model as a CKD model and aimed to evaluate the effects of HPD on kidney injury under conditions of non-CKD and CKD.

## 2. Results

### 2.1. HPD Loading Did Not Increase the Systolic Blood Pressure in Sham-Operated or Remnant Kidney Mice

The body weight (BW) gain of the 5/6 Nx + HPD group was significantly less than that of the sham + HPD or 5/6 Nx + ND group (Figure 1a). Furthermore, the systolic blood pressure (BP) of the 5/6 Nx groups was significantly higher compared with the sham groups (Figure 1b). However, HPD loading did not increase the systolic BP of the sham or 5/6 Nx groups (Figure 1b). In addition, the heart rates of all groups were identical during the experimental period (Figure 1c). As cardiac hypertrophy is closely associated with high BP, the heart weight/body weight ratio was also examined. The heart weight/body weight ratio was significantly higher in the 5/6 Nx groups than in the sham groups (Figure 1d). However, HPD loading did not increase the heart weight/body weight ratio in the sham or 5/6 Nx groups (Figure 1d).

### 2.2. HPD Loading Increased the Creatinine Clearance in Sham-Operated Mice and Urinary Albumin Excretion in Remnant Kidney Mice

Next, we assessed the kidney function of the mice by measuring the plasma creatinine and blood urea nitrogen (BUN) concentrations. The plasma creatinine concentration of the 5/6 Nx groups was significantly higher than that of the sham groups. However, HPD loading did not increase the plasma creatinine concentration in the sham or the 5/6 Nx groups (Figure 2a). Similarly, the BUN of the 5/6 Nx groups was significantly higher than that of the sham groups. However, HPD loading did not increase the BUN in the sham or the 5/6 Nx groups (Figure 2b). The creatinine clearance of the 5/6 Nx groups was significantly lower than that of the sham groups. However, HPD loading increased the creatinine clearance only in the sham groups (Figure 2c). Furthermore, urinary albumin excretion was significantly higher in the 5/6 Nx + HPD group than in the sham + HPD and 5/6 Nx + ND groups (Figure 2d).

### 2.3. HPD Loading Exacerbated Glomerular Injury in Remnant Kidney Mice

Next, we analyzed the histology of periodic acid–Schiff (PAS)-stained kidney sections (Figure 3a). We found that glomerular injury was aggravated in the 5/6 Nx groups compared with the sham groups. In addition, HPD loading exacerbated glomerular injury only in the 5/6 Nx groups (Figure 3b). We observed that tubular injury was exacerbated in the 5/6 Nx groups compared with the sham groups. However, HPD loading did not exacerbate tubular injury in the sham or the 5/6 Nx groups (Figure 3c). With regards to glomerular hypertrophy, the glomerular area was increased in the 5/6 Nx groups compared with the sham groups. However, HPD loading did not increase the glomerular area in the sham or the 5/6 Nx groups (Figure 3d). We observed that tubular injury was exacerbated in the 5/6 Nx groups compared with the sham groups. However, HPD loading did not exacerbate tubular injury in the sham or the 5/6 Nx groups (Figure 3c). The kidney protein expression level of neutrophil gelatinase-associated lipocalin (NGAL) in the 5/6 Nx groups was significantly increased compared with the sham groups (Figure 3e). However, HPD loading did not increase the kidney protein expression level of NGAL in the sham or 5/6 Nx groups (Figure 3e).

### 2.4. HPD Loading Exacerbated Podocyte Injury in Remnant Kidney Mice

A decreased number of dachshund family transcription factor 1 (DACH1)-positive podocytes is associated with podocyte injury [32,33]. Therefore, we examined the number of DACH1-positive podocytes in kidney sections as an alternative marker for podocyte injury (Figure 4a). The number of DACH1-positive podocytes decreased significantly in the 5/6 Nx groups compared with the sham groups (Figure 4b). In addition, HPD loading decreased the number of DACH1-positive podocytes only in the 5/6 Nx groups (Figure 4b).

### 2.5. HPD Loading Did Not Exacerbate Tubulointerstitial Fibrosis and Fibrosis-Related Genes in Sham-Operated or Remnant Kidney Mice

Next, we assessed the extent of tubulointerstitial fibrosis by examining picrosirius red (PSR)-stained kidney sections (Figure 5a). Tubulointerstitial fibrosis was exacerbated significantly in the 5/6 Nx groups compared with the sham groups (Figure 5b). However, HPD loading did not aggravate tubulointerstitial fibrosis in the sham or 5/6 Nx groups (Figure 5b). The kidney mRNA expression level of collagen type I in the 5/6 Nx groups was significantly increased compared with the sham groups (Figure 5c). However, HPD loading did not increase the kidney mRNA expression level of collagen type I in the sham or 5/6 Nx groups (Figure 5c).

## 3. Discussion

In the present study, the mice underwent sham surgery or 5/6 Nx, followed by HPD loading for 12 weeks. HPD loading increased urinary albumin excretion and aggravated glomerular and podocyte injury in the 5/6 Nx group. However, HPD loading did not lead to higher albumin excretion, glomerulosclerosis or podocyte injury or tubulointerstitial fibrosis in the sham-operated group. These results indicate that long-term HPD loading has little effect on the healthy kidney, while it may exacerbate glomerular and podocyte injury in a CKD condition. This is consistent with the results of a small randomized controlled trial, which suggested that an HPD intake did not affect kidney function in subjects with normal kidney function [34].

In the present study, we employed 129/Sv strain mice for the remnant kidney model. Although mice are often used to create such experimental models, including the 5/6 Nx model, they are resistant to kidney injury [35,36,37]. Additionally, their susceptibility to kidney injury is strain-dependent; for example, the 129/Sv strain is relatively susceptible to kidney damage compared with the C57BL/6 strain [38]. In addition, the 129/Sv strain develops hypertension in the 5/6 Nx model. Patients with CKD are typically also hypertensive [39,40,41]. Therefore, the selection of the 129/Sv strain in the 5/6 Nx mouse model could accurately reproduce the clinical features of patients with CKD.

In the present study, HPD loading did not increase systolic BP in the sham or 5/6 Nx groups. A cross-sectional study in humans reported that increased vegetable protein intake correlated with lower blood pressure [42]. In contrast, increased animal protein had no effect on the changes in blood pressure [42]. In the present study, most of the composition of the HPD was animal protein, which supports the hypothesis that animal protein has little effect on blood pressure.

In the present study, we hypothesized that long-term HPD loading was likely to aggravate kidney injury under the CKD condition of the remnant kidney model but would elicit little kidney injury in the healthy kidney of the sham-operated group. These differences in kidney injury suggest that HPD loading has different mechanisms of kidney injury between CKD and non-CKD conditions. HPD loading reportedly increased GFR under both CKD and healthy kidney conditions in the early stage [26,43]. Therefore, it is difficult to explain the difference in glomerular injury in response to HPD loading between the sham-operated and 5/6 Nx groups in the present study using only a hyperfiltration mechanism. We speculate that other mechanisms may be involved in the induction of kidney injury by HPD loading. As a possible mechanism of HPD loading-mediated exacerbation of kidney injury in the remnant kidney model, the increase in uremic toxins may be involved in the development of glomerular injury independent of hyperfiltration. Uremic toxins such as indoxyl sulfate are predominantly higher in CKD than in healthy kidney conditions [44]. The microbiome is altered in the CKD condition, and uremic toxins such as indoxyl sulfate produced by HPD loading are increased [45,46]. It has been reported that the increase in indoxyl sulfate can directly cause podocyte injury [47]. In the present study, 12 weeks of HPD loading provoked podocyte injury under the CKD condition in the remnant kidney model but not under the healthy kidney condition of the sham-operated kidney.

HPD loading did not exacerbate the tubular injury and interstitial fibrosis compared with the ND group under the CKD condition of the remnant kidney model, nor in the healthy kidney condition of the sham-operated groups. Under the CKD condition, the decrease in residual nephrons can lead to glomerular injury, followed by tubular injury. As a potential mechanism of tubular injury, protein leakage from the glomerulus exacerbates tubular injury [48]. In a study conducted using an adriamycin nephropathy model, it was found that under the conditions of severe proteinuria, little tubular injury was observed at 6 weeks and tubular injury was only observed at 16 weeks [49]. This previous study suggested that a prolonged loading period was necessary to exacerbate tubular injury, even in severe proteinuria. In the present study, the HPD-loading duration in the remnant kidney model may have been insufficient to exacerbate tubular injury and interstitial fibrosis.

The present study has some limitations. First, relatively few animals were studied. Second, uremic toxins such as indoxyl sulfate were not measured. Therefore, further studies are needed to elucidate the mechanism associated with HPD loading and accelerated glomerular injury under CKD conditions.

In conclusion, our findings suggest that HPD loading results in little or no kidney injury under healthy kidney conditions in sham-operated kidneys. However, HPD loading exacerbates glomerular injury under CKD conditions in a remnant kidney model. These results suggest that patients with CKD should follow a protein-restricted diet in order to prevent the exacerbation of kidney injury, while healthy individuals can maintain a high-protein diet without worrying about its adverse effects on the kidneys. Adequate protein intake has many benefits: (i) it maintains skeletal muscle mass and reduces sarcopenia in the elderly; (ii) it has positive effects on bone homeostasis and reduces fracture risk in the elderly; (iii) it promotes weight loss in obese patients by increasing satiety and energy expenditure, leading to insulin sensitivity. Given the results of the present study together with previous reports of the benefits of an HPD, overweight and elderly individuals without CKD may actively consume an HPD to maintain and improve their health.

## 4. Materials and Methods

### 4.1. Animals

All animal experiments were reviewed and approved by the Animal Studies Committee of Yokohama City University, which is in compliance with the ARRIVE guidelines. Efforts were made to minimize the number of animals used and ensure minimal suffering. Male 129/Sv mice were purchased from Japan SLC (Shizuoka, Japan). They were housed in a controlled environment on a 12-h light/dark cycle at a temperature of 25 °C and were allowed free access to an ND (24 % of total calories from protein, 59% carbohydrate, and 17% fat; Clea, Osaka, Japan) or HPD (45% protein, 38% carbohydrate and 17% fat; Clea, Osaka, Japan) and water.

### 4.2. Methods of 5/6 Nephrectomy and High-Protein Diet Loading

Male mice (8–11 weeks old) were assigned to the remnant kidney or sham-operated group. Only male mice were used in this study in order to limit the number of animals used and the variability associated with hormonal issues, which play a critical role in kidney injury. Two weeks after the surgery, the mice were continued on the ND or switched to an HPD for 12 weeks. Mice were divided into four groups: sham + ND (n = 5), sham + HPD (n = 4), 5/6 Nx + ND (n = 6) and 5/6 Nx + HPD (n = 6) groups. For the remnant kidney group, a right subcapsular Nx was performed, followed by surgical resection of the upper and lower thirds of the left kidney under isoflurane anesthesia, as described previously [50]. Subcapsular nephrectomy involves removal of the kidney after dissection of the renal capsule. The detailed surgical procedure is as follows. After placing the nose in the face mask for anesthesia, the forelegs and hind legs were fixed in dorsal recumbence on the heating pad with tape. After the abdominal skin was disinfected with 70% ethanol, a transverse incision was made to open the abdominal cavity. After identifying the right kidney and tearing the renal capsule entirely, the renal artery, vein and the right ureter were ligated at the renal hilum using a 4-0 silk thread. The right kidney was removed at the distal site of the ligation so as to not include the right adrenal gland. After identifying the left kidney and tearing the renal capsule entirely, the upper and lower thirds of the left kidney were ligated with a 4-0 silk thread and removed with scissors. Finally, the abdominal wall was closed with a 5-0 nylon suture. The surgical procedures were performed by a single experienced operator in order to ensure reproducibility.

### 4.3. BP Measurement

Systolic BP was measured using the tail-cuff method (BP-Monitor MK-2000; Muromachi Kikai Co., Tokyo, Japan), as described previously [51,52], between 09:00 and 14:00. At least 10 measurements were performed on each mouse, and the mean values were analyzed.

### 4.4. Biochemical Analysis

After mice were under deep anesthesia and unresponsive to all stimuli by inhaling 5% isoflurane anesthesia, blood samples were collected in the fed state by cardiac punctures. Experimental animals were killed humanely after anesthesia. Whole blood samples were centrifuged at 3000 rpm (MR-150, Tomy Seiko Co., Ltd., Tokyo, Japan) at 4 °C for 10 min to separate the plasma, which was stored at −80 °C until use. Plasma creatinine, BUN, urinary creatinine and urinary albumin concentrations were measured using a Hitachi 7180 autoanalyzer (Hitachi, Tokyo, Japan).

### 4.5. Metabolic Cage Analysis

Urine samples were collected by housing mice in metabolic cages, as described previously [53,54]. The mice had free access to tap water and were fed an ND or HPD.

### 4.6. Histological Analysis

Histological analyses were performed as described previously [55,56]. Briefly, mouse kidneys were fixed in 4% paraformaldehyde in PBS, incubated overnight at 4 °C and embedded in paraffin. Sections (4 μm thick) were stained with periodic acid–Schiff (PAS) and PSR.

Matrix accumulation and sclerosis of the glomerular tuft (glomerular sclerosis index) were determined in PAS-stained sections based on the following scoring system: 0—normal glomerulus; 1—mesangial expansion or sclerosis involving up to 25% of the glomerular tuft; 2—sclerosis of 25% to 50%; 3—sclerosis of 50% to 75% and/or segmental extracapillary fibrosis or proliferation; and 4—global sclerosis > 75%, global extracapillary fibrosis or complete collapse of the glomerular tuft [57]. The markers of tubular injury were scored based on the percentage of tubules in the corticomedullary junction that displayed tubular dilation, tubular atrophy, tubular cast formation, vacuolization, degeneration, sloughing off of tubular epithelial cells or loss of the brush border, and thickening of the tubular basement membrane as follows [58,59]: 0—none; 1—≤10%; 2—11–25%; 3—26–45%; 4—46–75%; and 5—>76%. The kidney sections were evaluated by nephrologists in an open-label fashion.

The glomerular area was measured by tracing the outline of the glomerular tuft of 25 glomeruli in the cortical fields of PAS-stained specimens. Fibrotic areas were measured digitally using a fluorescence microscope (BZ-X800; Keyence, Osaka, Japan) in the cortical fields of PSR-stained specimens. Immunohistochemistry was performed as described previously [60]. Sections were incubated with anti-DACH1 antibodies (1:200; 10914-1-AP; Proteintech, Wuhan, China). Immunohistochemical assays with primary antibodies were performed in tandem with specimens not in contact with the primary antibodies (negative control) (Appendix A). The number of DACH1-positive podocytes per glomerulus in a total of 30 glomeruli was counted in the cortical fields (magnification: ×200).

### 4.7. Real-Time Quantitative Reverse Transcription Polymerase Chain Reaction (PCR) Analysis

Total RNA was extracted from kidney tissues using ISOGEN (Nippon Gene, Tokyo, Japan), and cDNA was synthesized using the SuperScript III First-Strand System (Invitrogen, Carlsbad, CA, USA), according to the manufacturer’s protocol. Real-time quantitative reverse-transcription PCR analysis was performed using an ABI PRISM 7000 Sequence Detection System by incubating the reverse-transcription products with the TaqMan PCR Master Mix and designed TaqMan probes (Applied Biosystems, Foster City, CA, USA). The TaqMan probes used for PCR were collagen type I (col1a1), Mm00801666_g1. The mRNA levels were normalized to that of the 18S rRNA control.

### 4.8. Western Blotting Analysis

Protein expression was analyzed by Western blotting using tissue homogenates, as described previously [61,62]. Briefly, total protein extract was prepared from tissues with a sodium dodecyl sulfate-containing sample buffer. The protein concentration of each sample was measured with a NanoDrop One (Thermo Fisher Scientific, Waltham, MA, USA), using bovine serum albumin as the standard. Equal amounts of protein extract from each tissue sample (kidney tissue: 10 μg) were fractionated on a 5–20% polyacrylamide gel (Atto, Tokyo, Japan). The separated proteins were then transferred to a polyvinylidene difluoride membrane using the iBlot dry blotting system (Invitrogen). Membranes were blocked for 1 h at room temperature with phosphate-buffered saline containing 5% skim milk powder. Membranes were incubated with primary antibodies for NGAL (AF1857 1:800, R and D Systems, Minneapolis, MN, USA) and β-actin (A5441 1:5000, Sigma-Aldrich, St. Louis, MO, USA). Membranes were washed and then incubated with secondary antibodies for 60 min at room temperature. The sites of antibody–antigen reactions were visualized by enhanced chemiluminescence substrate (Merck, Kenilworth, NJ, USA). Images were analyzed quantitatively using a ChemiDoc Touch (Bio Rad, Hercules, CA, USA). Full unedited gel was shown (Appendix A).

### 4.9. Statistical Analysis

Statistical analysis was performed using GraphPad Prism software (GraphPad Software, La Jolla, CA, USA). All quantitative data are expressed as mean ± SEM. Differences between the four groups in multiple comparisons were assessed using a two-way repeated-measures analysis of variance (ANOVA) with Bonferroni’s post-hoc test for repeated measures data over time including BW, BP and HR. A two-way ANOVA with Tukey’s post-hoc test was used for the rest of the data.

## Figures and Tables

**Figure 1 ijms-24-07778-f001:**
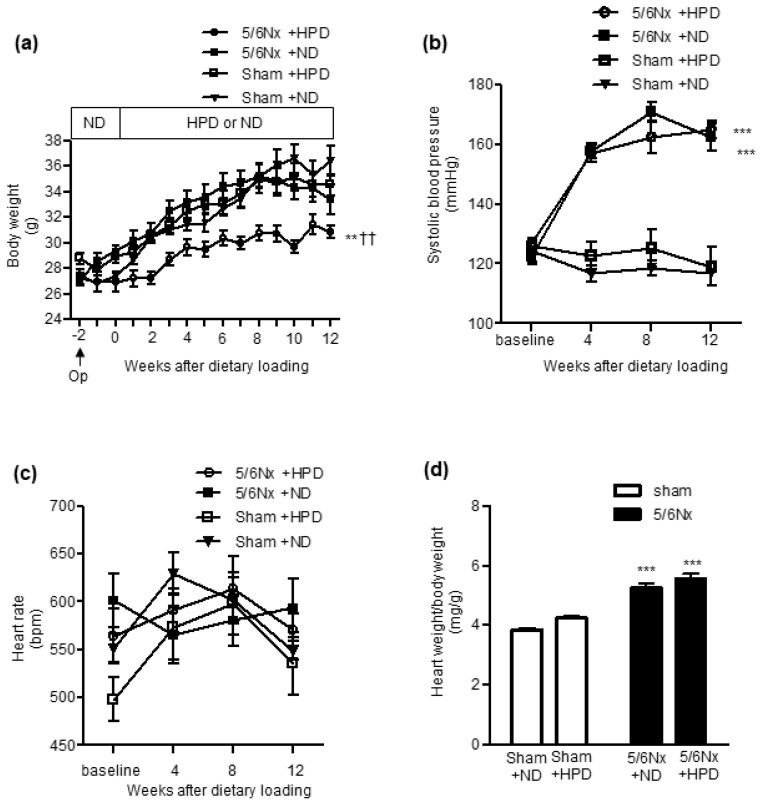
Effects of sham operation and 5/6 nephrectomy followed by high-protein diet loading on body weight, blood pressure, heart rate and heart weight/body weight ratio. (**a**) Body weight changes during normal diet (ND) or high-protein diet (HPD) loading in the sham operation (sham) and 5/6 nephrectomy (Nx) groups. (**b**) Systolic blood pressure and (**c**) heart rate in the sham + ND, sham + HPD, 5/6 Nx + ND and 5/6 Nx + HPD groups at baseline and after 4, 8 and 12 weeks of ND or HPD loading. Data are expressed as mean ± SEM (n = 4–6 per group). ** *p* < 0.01, *** *p* < 0.001 vs. sham group; †† *p* < 0.01 vs. ND group (two-way repeated measures ANOVA with Bonferroni’s post-hoc test). (**d**) Heart weight/body weight ratio of the sham + ND, sham + HPD, 5/6 Nx + ND and 5/6 Nx + HPD groups. Data are expressed as mean ± SEM (n = 4–6 per group). *** *p* < 0.001 vs. sham group (two-factorial analysis of variance and Bonferroni’s post-hoc analysis).

**Figure 2 ijms-24-07778-f002:**
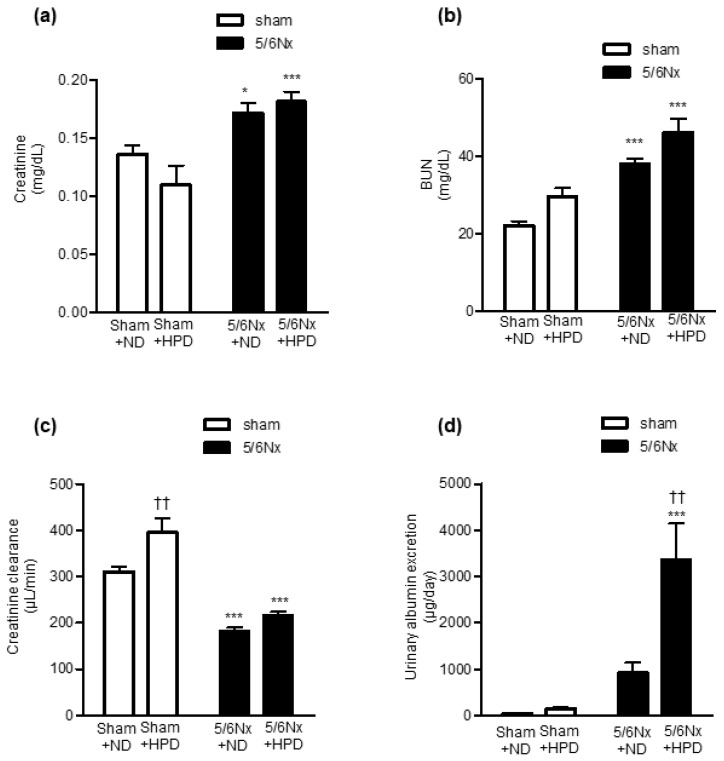
Effects of sham operation and 5/6 nephrectomy followed by high-protein diet loading on plasma creatinine, blood urea nitrogen, creatinine clearance and albuminuria. (**a**) Plasma creatinine concentration, (**b**) blood urea nitrogen (BUN), (**c**) creatinine clearance and (**d**) albuminuria in the sham operation (sham) + normal diet (ND), sham + high-protein diet (HPD), 5/6 nephrectomy (Nx) + ND and 5/6 Nx + HPD groups. Data are expressed as mean ± SEM (n = 4–6 per group). * *p* < 0.05, *** *p* < 0.001 vs. sham group; †† *p* < 0.01 vs. ND group (two-factorial analysis of variance and Bonferroni’s post-hoc analysis).

**Figure 3 ijms-24-07778-f003:**
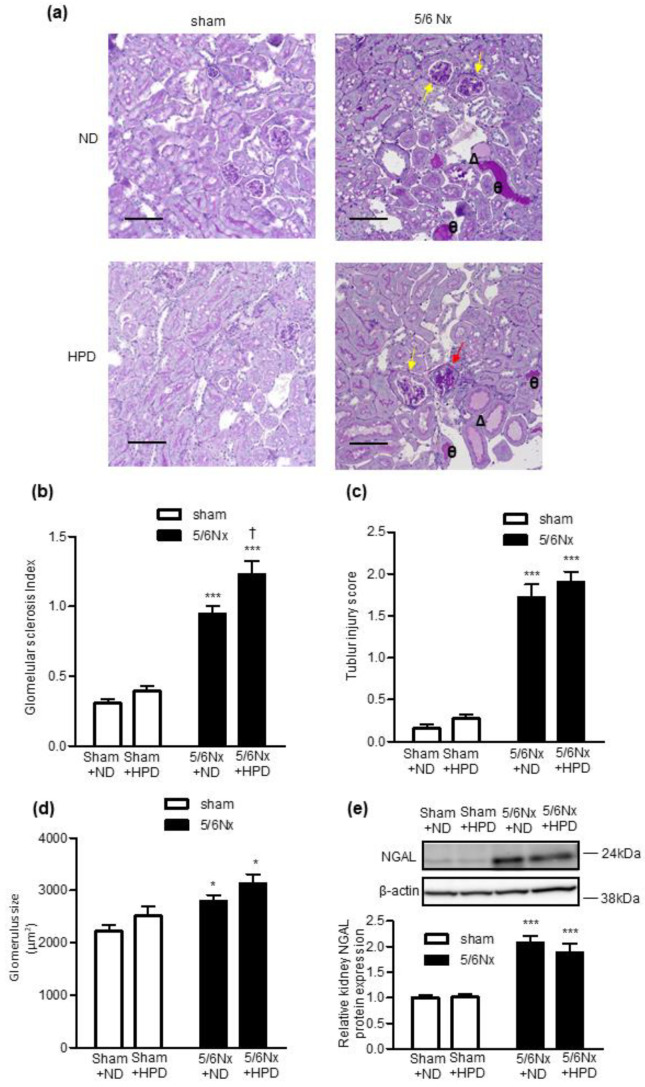
Effects of sham operation and 5/6 nephrectomy followed by high-protein diet loading on glomerular injury, tubular injury and glomerular size. (**a**) Representative images of periodic acid–Schiff-staining in kidney sections. Bar = 100 μm. Red arrows indicate glomerulosclerosis; yellow arrows indicate mesangial expansion; θ indicates urinary casts; Δ indicates tubular dilation. (**b**) Glomerular sclerosis index, (**c**) tubular injury score, (**d**) glomerular size and (**e**) relative kidney protein expression of neutrophil gelatinase-associated lipocalin (NGAL) in the sham operation (sham) + normal diet (ND), sham + high-protein diet (HPD), 5/6 nephrectomy (Nx) + ND, and 5/6 Nx + HPD groups. Data are expressed as mean ± SEM (n = 4–6 per group). * *p* < 0.05, *** *p* < 0.001 vs. sham group; † *p* < 0.05 vs. ND group (two-factorial analysis of variance and Bonferroni’s post-hoc analysis).

**Figure 4 ijms-24-07778-f004:**
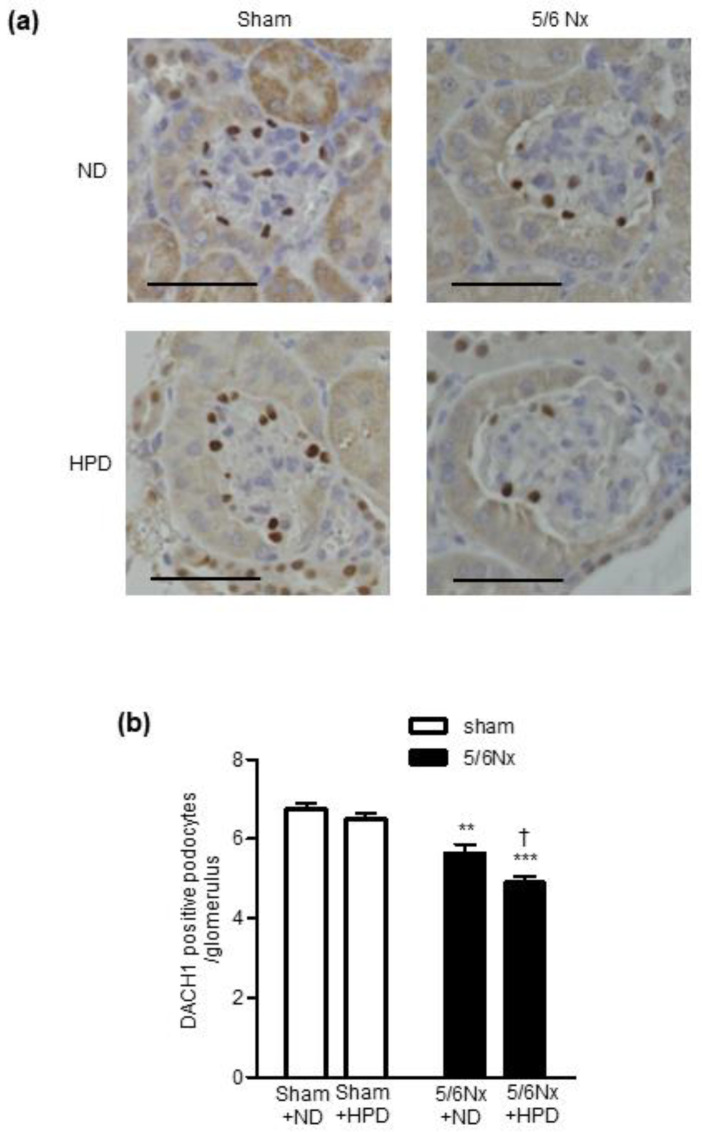
Effects of sham operation and 5/6 nephrectomy followed by high-protein diet loading on the number of dachshund family transcription factor 1 (DACH1)-positive podocytes. (**a**) Representative images of immunostaining for DACH1 in kidney sections. Scale bar = 50 µm. (**b**) The number of DACH1-positive podocytes in the sham operation (sham) + normal diet (ND), sham + high-protein diet (HPD), 5/6 nephrectomy (Nx) + ND, and 5/6 Nx + HPD groups. Data are expressed as mean ± SEM (n = 4–6 per group). ** *p* < 0.01, *** *p* < 0.001 vs. sham group; † *p* < 0.05 vs. ND group (two-factorial analysis of variance and Bonferroni’s post-hoc analysis).

**Figure 5 ijms-24-07778-f005:**
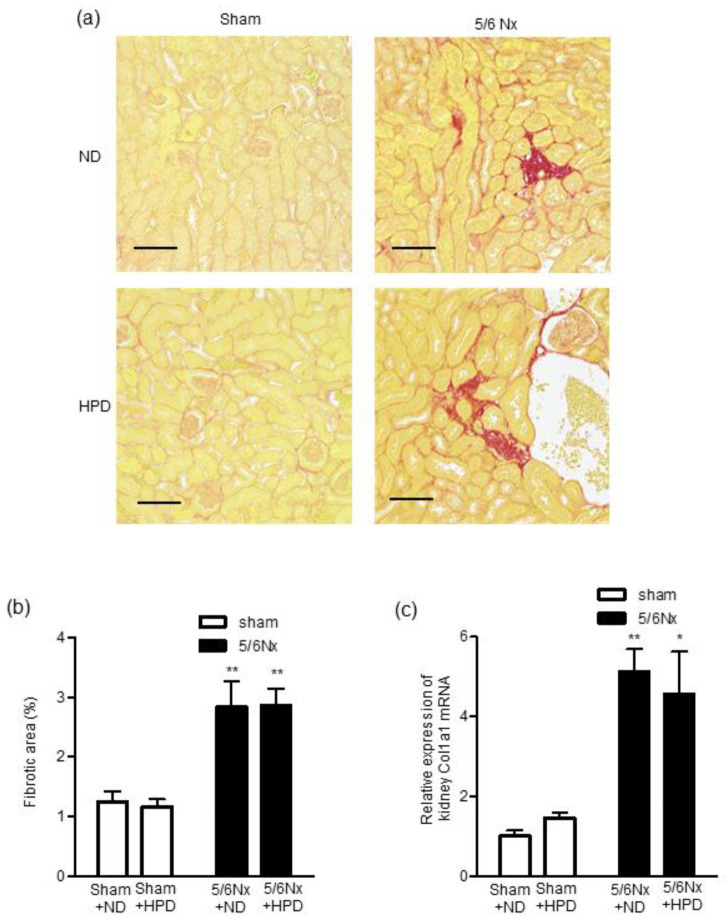
Effects of sham operation and 5/6 nephrectomy followed by high-protein diet loading on the severity of tubulointerstitial fibrosis and relative kidney mRNA expression of col1a1. (**a**) Representative images of picrosirius red-stained kidney sections. Bar = 100 μm. (**b**) Quantitative analysis of kidney fibrotic area and (**c**) relative kidney mRNA expression of col1al in the sham operation (sham) + normal diet (ND), sham + high-protein diet (HPD), 5/6 nephrectomy (Nx) + ND, and 5/6 Nx + HPD groups. Data are expressed as mean ± SEM (n = 4–6 per group). * *p* < 0.05, ** *p* < 0.01 vs. sham group (two-factorial analysis of variance and Bonferroni’s post-hoc analysis).

## Data Availability

The data used to support the findings of this study are included within the article.

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
