# Peer review of "Effects of a High-Protein Diet on Kidney Injury under Conditions of Non-CKD or CKD in Mice"

_ijms, 2023, doi:10.3390/ijms24097778_

Round 1
Reviewer 1 Report
Very nice and straightforward work. Excellent paper. Thanks. I only have two problems.
1. Lines 159 through 164 need to be reformatted and figure 5a is blocking part of the text.
2. At the very end of the Reference section, after reference number 44, there is a number 1 just sitting there for no reason.
Author Response
Response to the Comments of Reviewer 1
Very nice and straightforward work. Excellent paper. Thanks. I only have two problems.
Thank you very much for your favorable review. We appreciate your comments on our manuscript. We have revised the manuscript in compliance with your suggestions, and hope you agree with our revisions.
- Lines 159 through 164 need to be reformatted and figure 5a is blocking part of the text.
We appreciate your comments. According to your suggestion, we have revised the position of Figure 5a so that it does not block the text.
- At the very end of the Reference section, after reference number 44, there is a number 1 just sitting there for no reason.
Thank you very much for your comments. We have corrected the mistake.
Reviewer 2 Report
“Effects of a high-protein diet on kidney injury under conditions of non-CKD or CKD in mice” di Tanaka and colleagues aimed at investigating the effects of a high-protein diet (HPD) on kidney injury in a murine model with chronic kidney disease (CKD).
Although the topic is of current interest to the scientific community, the manuscript needs major revisions before publication in IJMS.
1. Abstract – This section is of fundamental importance because it should entice the reader to read the work in its entirety. In this case, although the authors have described the content of their manuscript in detail, in my opinion it is incomplete. In fact, it would be appropriate to add a concluding sentence suggesting possible future perspectives that might emerge from their findings.
2. In my opinion, the introduction is rather meagre and should necessarily be implemented. Already in the first sentence, the authors argue that the maintenance of optimal health during normal growth and during ageing occurs through adequate consumption of dietary protein. However, this aspect has not been explored in depth, although there is plenty of evidence for it in the literature. The authors should also explain what is meant by a high-protein diet for the reader's better understanding. In addition, I would also suggest that the introduction be expanded upon with some mention of the CDK epidemiology.
3. How many animals were used in this study? Please specify this in the methods section.
4. The authors should provide negative controls for the staining performed, perhaps in a supplementary section.
5. The histological analysis performed does not provide an appropriate quantitative evaluation to support these results. The authors should implement their results by investigating the expression of renal damage markers by western blotting analysis.
6. In the concluding section, it is important that this section explicitly outlines the added value of this study and what innovation it provides to the current literature.
Author Response
Response to the Comments of Reviewer 2
“Effects of a high-protein diet on kidney injury under conditions of non-CKD or CKD in mice” di Tanaka and colleagues aimed at investigating the effects of a high-protein diet (HPD) on kidney injury in a murine model with chronic kidney disease (CKD).
Although the topic is of current interest to the scientific community, the manuscript needs major revisions before publication in IJMS.
- Abstract – This section is of fundamental importance because it should entice the reader to read the work in its entirety. In this case, although the authors have described the content of their manuscript in detail, in my opinion it is incomplete. In fact, it would be appropriate to add a concluding sentence suggesting possible future perspectives that might emerge from their findings.
We appreciate and agree with your comments. According to your suggestion, we have revised the Abstract section as follows: "Clinical application of these results suggests that patients with CKD should follow protein-restricted diet to prevent the exacerbation of kidney injury, while healthy individuals can have HPD without worrying about the adverse effects (Abstract; page 1, line 38-40)".
- In my opinion, the introduction is rather meagre and should necessarily be implemented. Already in the first sentence, the authors argue that the maintenance of optimal health during normal growth and during ageing occurs through adequate consumption of dietary protein. However, this aspect has not been explored in depth, although there is plenty of evidence for it in the literature. The authors should also explain what is meant by a high-protein diet for the reader's better understanding. In addition, I would also suggest that the introduction be expanded upon with some mention of the CDK epidemiology.
We appreciate your comments. Comply with your comments, we have intensively revised the Introduction section as follows: "Adequate consumption of dietary protein is critical for the maintenance of optimal health during normal growth and aging. One of the effects of high-protein diet (HPD) intake is to promote muscle protein synthesis [1]. This effect inhibits the loss of skeletal muscle mass due to sarcopenia in the elderly [2,3]. Furthermore, HPD intake positively affects calcium and bone homeostasis through its effects on calcium absorption, bone turnover [4,5]. A significant positive association between protein intake and born mineral density has been shown by epidemiologic studies [6]. HPD intake could inhibit age-related bone loss and decrease fracture risk due to osteoporosis [7,8]. In addition, HPD intake has effects of lower daily energy intake through increased satiety [9] and higher daily energy expenditure through the increased thermic effect [10]. Therefore, HPD intake decreases body weight and maintains lean body mass in obese patients [11,12]. Furthermore, protein intake regulates endogenous glucose metabolism and insulin sensitivity [13,14]. It has been shown that HPD intake stabilized blood glucose and lowered postabsorptive and postprandial insulin secretion during weight loss [15]. Cross-sectional clinical studies showed that dairy protein consumption was inversely related to the incidence of impaired fasting glycemia and type 2 diabetes [16,17]. Considering the prevalence of overweight individuals, obesity, and global aging, the consumption of a HPD may be advantageous.
Although HPD has many benefits, we must also consider the effects of a high-protein diet on chronic kidney disease (CKD). The number of patients with CKD continues to increase worldwide. The progression of CKD culminates with end-stage kidney disease, which eventually necessitates kidney replacement therapy [18–20]. Patients with CKD are at a high risk of mortality due to a high incidence of cardiovascular disease that is associated with kidney disease [21,22]. In 2017, 697.5 million cases of CKD were reported worldwide, with 1.2 million people ultimately dying from the disease [23]. Therefore, the development of interventions that prevent the progression of CKD represents a global challenge. (Introduction; page 2, line 46-72)".
- How many animals were used in this study? Please specify this in the methods section.
We appreciate your comments. According to your suggestion, we have revised the Material and Methods section as follows: " Mice were divided into four groups: sham + ND (n = 5), sham + HPD (n = 4), 5/6 Nx + ND (n = 6), and 5/6 Nx + HPD (n = 6) groups (Material and Methods; page 10, line 300-302)".
- The authors should provide negative controls for the staining performed, perhaps in a supplementary section.
We appreciate your comments and understand the concerns. Comply with your comments, we have performed additional experiments to examine the negative control for the staining immunohistochemistry using the anti-Dachshund Family Transcription Factor 1 antibody (new Supplementary Figure 1). We have revised the Materials and Methods section as follows: "Immunohistochemical assays with primary antibodies were performed in tandem with specimens not in contact with the primary antibodies (negative control) (Supplementary Figure 1) (Materials and Methods; page 11, line 358-360)".
- The histological analysis performed does not provide an appropriate quantitative evaluation to support these results. The authors should implement their results by investigating the expression of renal damage markers by western blotting analysis.
We appreciate your comments and understand the concerns. According to your suggestion, we have performed additional experiments to examine the kidney protein expression level of neutrophil gelatinase-associated lipocalin (NGAL) (new Figure 3e). We have revised the Material and Methods section as follows: "4.8. Western blotting analysis: Protein expression was analyzed by western blotting using tissue homogenates, as described previously [45,46]. Briefly, total protein extract was prepared from tissues with sodium dodecyl sulfate-containing sample buffer. The protein concentration of each sample was measured with NanoDrop One (Thermo Fisher Scientific), using bovine serum albumin as the standard. Equal amounts of protein extract from each tissue samples (kidney tissue: 10 μg) were fractionated on a 5–20% polyacrylamide gel (Atto, Tokyo, Japan). The separated proteins were then transferred to a polyvinylidene difluoride membrane using the iBlot Dry Blotting System (Invitrogen). Membranes were blocked for 1 h at room temperature with phosphate-buffered saline containing 5% skim milk powder. Membranes were incubated with primary antibodies for NGAL (AF1857 1:800, R and D SYSTEMS, Minneapolis, MN, USA) and β-actin (A5441 1:5,000, Sigma-Aldrich, St. Louis, MO, USA). Membranes were washed and then incubated with secondary antibodies for 60 min at room temperature. The sites of antibody–antigen reactions were visualized by enhanced chemiluminescence substrate (Merck, Kenilworth, NJ, USA). Images were analyzed quantitatively using a ChemiDoc Touch (Bio Rad, Hercules, CA, USA) (Material and Methods; page 12, line 378-394)". Additionally, we have revised the Results section as follows: “The kidney protein expression level of neutrophil gelatinase-associated lipocalin (NGAL) in the 5/6 Nx groups was significantly increased compared with the sham groups (Fig. 3e). However, HPD loading did not increase the kidney protein expression level of NGAL in the sham or 5/6 Nx groups (Fig. 3e) (Results; page 5, line 159-162).”
- In the concluding section, it is important that this section explicitly outlines the added value of this study and what innovation it provides to the current literature.
We appreciate the reviewer`s comments and understand the concerns. We have revised the Discussion section as follows: "In conclusion, our findings suggest that HPD loading results in little or no kidney injury under healthy kidney conditions in sham-operated kidneys. However, HPD loading exacerbates glomerular injury under CKD conditions in a remnant kidney model. These results suggest that patients with CKD should follow a protein-restricted diet in order to prevent the exacerbation of kidney injury, while healthy individuals can have a high-protein diet without worrying about its adverse effects on the kidneys. Adequate protein intake has many benefits: 1) it maintains skeletal muscle mass and reduces sarcopenia in the elderly; 2) it has positive effects on bone homeostasis and reduces fracture risk in the elderly; 3) it promotes weight loss in obese patients by increasing satiety and energy expenditure, leading to insulin sensitivity. Given the results of the present study together with previous reports of the benefits of HPD, overweight and elderly individuals without CKD may actively consume HPD to maintain and improve their health (Discussion; page 10, line 277-283)".
Round 2
Reviewer 2 Report
The authors met my demands and the manuscript improved considerably. Therefore, it is suitable for publication in IJMS.